# Indoor Positioning Using Magnetic Fingerprint Map Captured by Magnetic Sensor Array

**DOI:** 10.3390/s21175707

**Published:** 2021-08-24

**Authors:** Ching-Han Chen, Pi-Wei Chen, Pi-Jhong Chen, Tzung-Hsin Liu

**Affiliations:** 1Department of Computer Science and Information Engineering, National Central University, Taoyuan 32001, Taiwan; pierre@g.ncu.edu.tw (C.-H.C.); nickyliu6@gmail.com (T.-H.L.); 2Department of English, Wenzao Ursuline University of Languages, Kaohsiung 80793, Taiwan; 3Undergraduate Program in College of Electrical Engineering and Computer Science, Chung Yuan Christian University, Taoyuan 32023, Taiwan; s10720107@cycu.edu.tw

**Keywords:** magnetic field, indoor positioning, magnetic sensor array, recurrent probabilistic neural network

## Abstract

By collecting the magnetic field information of each spatial point, we can build a magnetic field fingerprint map. When the user is positioning, the magnetic field measured by the sensor is matched with the magnetic field fingerprint map to identify the user’s location. However, since the magnetic field is easily affected by external magnetic fields and magnetic storms, which can lead to “local temporal-spatial variation”, it is difficult to construct a stable and accurate magnetic field fingerprint map for indoor positioning. This research proposes a new magnetic indoor positioning method, which combines a magnetic sensor array composed of three magnetic sensors and a recurrent probabilistic neural network (RPNN) to realize a high-precision indoor positioning system. The magnetic sensor array can detect subtle magnetic anomalies and spatial variations to improve the stability and accuracy of magnetic field fingerprint maps, and the RPNN model is built for recognizing magnetic field fingerprint. We implement an embedded magnetic sensor array positioning system, which is evaluated in an experimental environment. Our method can reduce the noise caused by the spatial-temporal variation of the magnetic field, thus greatly improving the indoor positioning accuracy, reaching an average positioning accuracy of 0.78 m.

## 1. Introduction

Location-based service (LBS) is a value-added service that uses positioning technology to precisely provide user location information. Because this service can provide the most direct advertisements, weather information, according to the user’s current location, makes it of great commercial value. Typical positioning methods can be roughly divided into indoor and outdoor applications, according to the field of use. Global navigation satellite systems (GPS) are often implemented for outdoor positioning, but GPS signals are greatly attenuated when obstructed by obstacles, which makes it hard to apply to indoor positioning. Conversely, the commonly used indoor positioning solutions are based on the sensing devices used, such as Wi-Fi [1], Bluetooth [2], laser rangefinder [3], visual orientation [4], etc. The advantage of Wi-Fi/Bluetooth positioning technology is that the wireless positioning operation has a wide range, and the deployment can be arranged according to the needs of users, but the disadvantage is that the minimum cost of placement is high, and the interference between the object under test and the landmark is easy to cause positioning deviation [5,6]. The laser rangefinder positioning method is currently the most commonly used positioning method for the automatic navigation of indoor unmanned vehicles. The unmanned vehicle with the laser rangefinder can construct the map and achieve the purpose of coordinate positioning during the movement; the advantage is that the cost of deployment is low, and the positioning accuracy is high, but the disadvantage is that the cost of this technology is high. As for visual orientation, this technology uses cameras to measure the distance between objects, calculating their latent information; its advantage is having high accuracy, but the disadvantage is that the positioning range is limited by the camera’s viewing angle and distance, therefore it is not suitable for large-scale positioning tasks. Moreover, it is easily affected by light, causing it to perform unstably in conditions in which light changes drastically.

Compared with the above-mentioned traditional indoor positioning method, the magnetic field has the characteristic of no interference by the movement of a crowd and does not require additional hardware deployment, thus it has great potential for indoor positioning [7,8,9]. The magnetic field is an innate resource of the earth, but when it passes through the steel structure in the building and various electrical equipment each location in the room will have its own different magnetic field data. Therefore, as long as we use these unique magnetic field features to build a dataset (fingerprint map) in advance, we can use it for indoor positioning and navigation purposes. There are several papers published that also suggest this promising usage of magnetic field positioning, such as accurate magnetic indoor localization using deep learning [10], multi-view graph [11], and reliability-augmented particle filter [12]. The current magnetic field positioning technology mainly uses the fingerprinting method [10,11,12,13,14] by comparing each point in the room. This method needs to split the indoor environment into a grid and consequently collect the magnetic field information of each point. When the user is positioning, the magnetic field information collected by the sensor is matched with the information of the fingerprint map, such that the location of the user can be accurately found. However, since the magnetic field is easily affected by geomagnetic storms, the current magnetic field positioning technology is mostly combined with other indoor positioning technologies such as Wi-Fi [15] or inertial sensor (IMU) [12] to improve the accuracy.

This research proposes a high-performance indoor positioning system using a magnetic sensor array. We use three MMC5883MA magnetic sensors to build a magnetic sensor array and develop a low-complexity recurrent probabilistic neural network classifier [16] on the embedded platform “STM32F767ZI” to calculate the position information, and finally complete a high-accuracy, low-power consumption indoor positioning system. In this system, we use the Kalman filter to filter out the environmental noise of the magnetic field signal output by our magnetic sensor array and calculate the differentiation computation value between the sensors to reduce the influence of the time changing of the indoor magnetic field. This allows us to obtain stable and effective magnetic field characteristic information. Compared with other indoor positioning technologies, the classifier architecture used by the system also has the advantages of low-complexity and low-power consumption and is suitable for application in low-cost electronic products.

## 2. Related Work

### 2.1. Geomagnetic Field Measurement

According to the speculation of geomagnetic experts, the earth’s magnetic field may be formed by the flow of high-temperature metal fluid inside the earth. Therefore, we can imagine that there is a large magnet rod inside the earth. Geomagnetic poles are two points on the two sides of the earth, close to the geographic North and South, and the axis passing through the two points is called the geomagnetic axis. The geomagnetic axis does not coincide with the earth’s spin axis but has an angle of about 11 degrees. The commonly used compass points in the direction through which the geomagnetism passes. This direction is usually called magnetic north. Because the magnetic north is not the same as the geographic north pole, this deviation angle is regarded as the magnetic declination.

#### 2.1.1. Magnetic Field Strength

The earth’s magnetic field intensity (*F*) is a directional vector in space, which can be decomposed into three components *Mx*, *My*, and *Mz*, representing respectively to north, east, and vertically down (as shown in Figure 1a), and the magnetic field intensity *F* is as:(1)$$F=\sqrt{Mx^{2}+My^{2}+Mz^{2}}$$

#### 2.1.2. Geomagnetic Declination

Since the sensor cannot obtain the true north direction of the earth, we assume that when the X-axis of the sensor points to the true north, the magnetic declination (*D*) can be calculated through the angle between the horizontal component (*H*) of the earth’s magnetic field, and that of the true north direction. Figure 1b shows the geomagnetic declination *D* as
(2)$$D=arctan\frac{M_{y}}{M_{x}}$$

#### 2.1.3. Geomagnetic Inclination

The magnetic inclination (*I*) is the angle between the geomagnetic intensity vector and the horizontal component (as shown in Figure 1c). In this study, we use *Mx* and *My* to calculate the horizontal component (*H*) and use this as a benchmark to calculate the magnetic inclination (*I*). The calculation formula is shown in Formulas (3) and (4):(3)$$H=\sqrt{M_{x}{}^{2}+M_{y}{}^{2}}$$
(4)$$I=arctan\frac{M_{z}}{H}$$

### 2.2. Kalman Filter

The Kalman filter has long been regarded as the optimal solution for object tracking and data prediction. It can calculate the situation of a dynamic system from a series of incomplete and noise-containing measurements. Thus, it is often used to correct noisy time-series data [17]. Kalman filtering is a recursive estimation, that is, as long as knowing the estimated value of the previous moment and the observed value of the current state, the final estimated value of the current state can be calculated. Thus, there is no need to record observations or historically estimated information. The operation of the Kalman filter is divided into two states: prediction and measurement value update. In the case of predicting state, the estimated value of the previous moment will be used as a feature to make a prediction of the current state, and under the state of value updating, the filter uses the observed value of the current state to optimize the predicted value, such that we can obtain a more accurate new estimation.

#### 2.2.1. Predict Step

(5)$${\hat{X}}_{\bar{t}}=F{\hat{X}}_{\bar{t}-1}+Bu_{t-1}$$(6)$$P_{\bar{t}}=FP_{t-1}F^{T}+Q$$
where $\hat{X}$ is an estimate of *X*, *P* is called the state error covariance, *F* and *B* are the state transition matrix, and *Q* is the noise covariance matrix.

#### 2.2.2. Update Step

(7)$$k_{t}=P_{\bar{t}}H^{T}{\left(HP_{\bar{t}}H^{T}+R\right)}^{-1}$$(8)$${\hat{X}}_{t}={\hat{X}}_{\bar{t}}+k_{t}\left(Z_{t}-H{\hat{X}}_{\bar{t}}\right)$$(9)$$P_{t}=\left(I-K_{t}H\right)P_{\bar{t}}$$
where *k* is the Kalman gain, *H* is the observation matrix, *Z* is the observation matrix, and *R* is the noise covariance matrix of the observation.

### 2.3. Probabilistic Neural Network

A single spread probabilistic neural network classifier (PNN) is an implementation of a statistical algorithm. RPNN was first proposed in [18]. The architecture of a PNN consists of four layers, which includes the input layer, Gaussian layer, summation layer, and decision layer, as shown in Figure 2.

The input layer in a PNN is a linear combination of several multidimensional features. As for the Gaussian layer, the probabilistic density function (*pdf*) of each Gaussian neuron is as shown in the following formula:(10)$$p_{j}^{i}(X)=\frac{1}{{\left(2\pi \right)}^{n/2}\sigma^{n}}\exp(-\frac{{\left(X-Y_{j}^{i}\right)}^{T}(X-Y_{j}^{i})}{2\sigma^{2}})$$
where *n* is the indication of the length of the input vector *X*; *Y_j_^i^* indicates the *n*-dimensional *j*th training example from class *C_i_*, and the smoothing parameter is denoted as σ. The calculation of the average of pdf for *N_i_* training samples can be denoted as follows:(11)$$G{(C}_{i}\left|X\right.)=\frac{1}{{\left(2\pi \right)}^{n/2}\sigma^{n}}(\frac{1}{N_{i}})\sum_{j=1}^{N_{i}}\exp(-\frac{{\left(X-Y_{j}^{i}\right)}^{T}(X-Y_{j}^{i})}{2\sigma^{2}})$$
where *N_i_* denotes the total number of samples in class *C_i_*.

Once passed the summation layer, the input vector *X* will be summed to the output layer, which contains a single neuron. The goal of the single neuron is to give the classification decision for the input according to the Bayes decision rule. The function is as follows:(12)$$C(X)=\underset{i=1,2,..,m}{\arg\max}\left\{\frac{1}{{\left(2\pi \right)}^{n/2}\sigma^{n}}(\frac{1}{N_{i}})\sum_{j=1}^{N_{i}}\exp(-\frac{{\left(X-Y_{j}^{i}\right)}^{T}(X-Y_{j}^{i})}{2\sigma^{2}})\right\}$$
where the number of classes presented in the system is denoted as *m*.

### 2.4. Recurrent Probabilistic Neural Network

Recurrent probabilistic neural network (RPNN) was first proposed in [18]. It is a neural network that uses PNN as a core model and combines RNN and long short-term memory (LSTM) algorithms in one architecture, which allows the probabilistic neural network to have long-term and short-term memory functions.

Schematic diagram of the recurrent probabilistic neural network is shown in Figure 3. The equation of memory unit update procedure is shown in follow: Such as:(13)$$\begin{array}{l} p_{i\_update}(t)=(1-\delta )\cdot (p_{i\_update}(t-1)+P_{i}^{*}(t)),\;\mathrm{if}\;P_{i}(t)=\begin{array}{c} \max\\ i \end{array}P(t)\\ p_{i\_update}(t)=\delta \cdot (p_{i\_update}(t-1)+P_{i}^{*}(t)),\;\;\mathrm{otherwise} \end{array}$$
and
(14)$$P_{i}^{*}(t)=\left\{\begin{array}{c} K\cdot P_{i}(t),\;\mathrm{if}\;t\neq 0\\ 0,\;\;\mathrm{otherwise} \end{array}\right.$$
where *δ* is the forgetting factor of the memory unit. *K* is a parameter of the length of memory. *P*(*t*) is the probability of dividing in the *t*-th time, and *P_i_**(*t*) is the output value of the updated memory cell.

The parameter σ in RPNN is the Gaussian function smoothing coefficient. The smoothing coefficient σ determines the breadth of its distribution. The larger the σ^2^ value, the wider the distribution, the higher the noise can be tolerated. The smaller the σ^2^ value, the narrower the distribution, the lower the noise can be tolerated, and the σ is classified according to different classifications.

To make the neural network adaptive, we use the particle swarm optimization (PSO) algorithm to adjust the σ, δ, and K parameters in the recurrent neural unit to model the accuracy as a particle. The fitness function of the group optimization algorithm iterates the particles in the search space toward the optimal solution to obtain robust recognition performance.

## 3. Magnetic Sensor Array Indoor Positioning System

The indoor positioning system based on the magnetic sensor array described in this research is integrated with the STM32F767 Nucleo-144 development platform as shown in Figure 4. The magnetic sensor array system is used to collect the magnetic field information of the environment in order to calculate magnetic field features, such as magnetic field X/Y/Z components, magnetic field strength, magnetic inclination, and magnetic declination, etc. We also perform differentiation computation on the data measured by each sensor to obtain feature values that can reduce the time-varying influence.

The magnetic sensor array indoor positioning system is composed of three magnetic sensors MMC5883MA; the distance between the sensors is 5 cm, and the three sensors are arranged in an L-shape. The sensor will return the collected magnetic field information to the MCU STM32F767 Nucleo-144 development platform through the I2C communication interface for signal processing and magnetic field feature value calculation, and finally sent to the recurrent probabilistic neural network classifier to estimate the coordinates of the location. Figure 5 shows the design and prototype of the magnetic sensor array indoor positioning system.

### 3.1. Pre-Processing of Signals

The magnetic field information output by the MMC5883MA sensor will have certain noise that results in unstable changes in the output signal; therefore, after initializing the MMC5883MA Magnetic sensor, we will first use the Kalman filter to process the output of the X/Y/Z magnetic field components, which are collected by the sensors. Finally, we can obtain a more stable magnetic field output information.

Besides the magnetic field X/Y/Z components, magnetic field strength, magnetic inclination, and magnetic declination, and the magnetic field features mentioned above, we also include the differentiation computation values measured by three sensors in the sensor array: M1, M2, M3, in order to obtain the differentiation feature. For the X component, the calculation of differentiation includes Mx1−Mx2, Mx1−Mx3, and Mx2−Mx3; Y component are My1−My2, My1−My3, andMy2−My3; and Z component are Mz1−Mz2, Mz1−Mz3, and Mz2−Mz3. These form the magnetic feature vector at each location.

### 3.2. Build the Magnetic Field Database

Before conducting indoor positioning, we use the magnetic sensor array system to collect the magnetic field data of different indoor positions in a specific building and consequently establish a fingerprint database of the indoor magnetic field, which is used to form a fingerprint map. For magnetic field data collection, we divide the indoor space into grids, as shown in Figure 6, and then sequentially collect the magnetic field information of each coordinate point and save it in the database.

Figure 7 is a schematic diagram of using the magnetic sensor array system to collect the magnetic field data of the indoor space in Figure 6 and estimate the indoor magnetic field intensity distribution based on the information of the sampling point. From this experiment, it can be concluded that the indoor magnetic field in the building does verify from each space, creating a set of unique features due to the fixture of the building and the indoor facilities, and the range is about 30 to 60 µT. Therefore, the magnetic field strength of each indoor location can be regarded as a significant feature in the indoor environment and can be used in indoor positioning applications.

### 3.3. Estimation of Position Coordinates

We used RPNN to construct a recognition model for inferencing the position coordinates by magnetic fingerprint map. This model is shown in Figure 8, called RPNN fingerprint map classifier (RFMC). The input vector of the RFMC includes the magnetic field strength, magnetic declination, magnetic inclination, magnetic field X/Y/Z components, and their differentiation captured by the magnetic sensor array. The magnetic fingerprint map will be placed into RFMC. After being processed by several hidden layers such as Gaussian layer, summation layer, and decision layer, it will output the position coordinates as shown in Figure 8.

## 4. Experimental Results and Discussion

### 4.1. Experiment of Magnetic Field Feature

The earth’s magnetic field is affected by the sun’s radiation, which will cause the magnetic field information measured at different times to be different. Therefore, we must first conduct experiments on the stability of the magnetic field features that were used.

#### 4.1.1. Changes in Magnetic Field Strength at Different Times

We used the same magnetic sensor MMC5883MA to measure the magnetic information of a certain point in an indoor space at room temperature. We recorded once every 1 min from 8 pm on 4/22, 05/06, 05/20, for a total of 24 h, to collect the changes of magnetic field strength in the same environment within 14 days. As we can see in Figure 9, the waveform changes and morphology of magnetic field components X/Y/Z within 24 h are similar. The variation of the magnetic field intensity of the X-axis component was about 0.22 µT, and for the Y-axis component, it was about 0.25 µT. As for the Z-axis component, it was about 0.25 µT.

#### 4.1.2. Measurement from Magnetic Sensor Array System

In order to ensure that the data from different sampling points collected by the sensors were unique and suitable for the usage of input features for indoor positioning, we used the magnetic sensor array system to sample at the same point. The magnetic field data was recorded once every 1 min for a total of 24 h. Next, we conducted differentiation computation of the X/Y/Z components output by the three magnetic sensors M1, M2, M3 in the magnetic sensor array to verify the stability of the value of the signal. After the computation, the value of the X component after the differentiation computation was about 0.15 µT, that of the Y component was about 0.1 µT, and that of Z component after the calculation was about 0.2 µT; each component of the magnetic field intensity captured by the magnetic sensor array was displayed after the differentiation computation, and we can see the value was stable, as shown in Figure 10.

Next, we constructed a 4 × 4 grid area in a certain space of the building, with 16 sampling points in total; the length and width of the grid was 60 cm, as shown in Figure 11. Then we used the magnetic sensor array system proposed to collect the magnetic field information at each sampling point and performed the differentiation computation of the X/Y/Z components. The results are shown in Table 1. According to the data listed in Table 1, if the distance between two sampling points is only 60 cm, the differentiation calculations we used can remain stable and are distinguishable, which allows us to use it as a magnetic field feature.

### 4.2. Experiment of Sensor Array Positioning

#### 4.2.1. Simulate Environment

In this experiment, we used the corridor and hall in our laboratory as our experimental environment, as shown in Figure 12. The length of the laboratory corridor was 46 m, and magnetic field information was collected every 60 cm. Twenty data items were collected at each sampling point, for a total of 1540 data items. The length and width of the hall were 11.4 m and 6.6 m, respectively, and the total area was 75.24 square meters. We divided it into 20 × 12 grids and collected magnetic field information every 60 cm and collected 20 data at each sampling point. A total of 4800 magnetic fields were collected to build a fingerprint database.

#### 4.2.2. Configuration of RPNN Fingerprint Map Classifier

In the experimental environment described in Section 4.2.1, the day after we collected and established the magnetic fingerprint database, we recollected the magnetic field data at the same sampling points in the lobby and corridor of the building. Two data were collected at each coordinate point. As for the test data for indoor positioning, it was used to verify the accuracy of the RPNN fingerprint map classifier (RFMC) for location recognition. In order to confirm the optimal smoothing parameter of the RFMC, we selected the classifier models established with the smoothing parameters of 0.3, 0.2, and 0.1 to perform the position recognition experiment. The results are shown in Table 2. From the experimental results of Table 2, it can be seen that no matter where the test field is, the best result is under the condition that the smoothing parameter of the RFMC is set to less than 0.2. However, if the smoothing parameter is set to 0.1, the maximum error will be slightly increased compared to the setting of 0.2. Therefore, for the RFMC, the best choice of smoothing parameters is between 0.1 and 0.2.

#### 4.2.3. Comparison of Positioning Accuracy between Single Sensor and Sensor Array

In this experiment, we set a rectangular test path in the hall to compare the results of the indoor positioning accuracy of a single magnetic sensor and a magnetic sensor array. Figure 13 is the result of using a single sensor. The average error was 1.17 m, the maximum error reached 5.07 m.

When we used the X/Y/Z components, magnetic field strength, magnetic inclination angle, and magnetic declination angle collected by this sensor array as the input vector of the RFMC, the result performed better than that of using a single sensor. The mean error of the experimental results was 0.83 m, and the maximum error was 1.91 m as shown in Figure 14.

Figure 15 shows the inference result of RPNN fingerprint map classifier (RFMC) for experimental indoor positioning. This time, we added a new feature, the differentiation computation between the three sensors. In this testing experiment, the prediction result of the RFMC was almost identical to the authentic path. The average error of the experimental results was 0.73 m, and the maximum error was 1.73 m. It can be known from the above experiments that the magnetic sensor array in this research can definitely improve the positioning accuracy compared with the single magnetic sensor. In addition, if we impose the differentiation calculations to the information collected by the three sensors, the positioning accuracy can be boosted further as shown in Table 3.

### 4.3. The Comparison of the Accuracy of Different Positioning Methods

Compared with other methods of positioning, the method of using convolutional neural network (CNN) algorithms to classify hybrid location images, which use Wi-Fi and magnetic field fingerprints to establish, the mean error of the result was 1.7 m, and 90% of the error was maintained within 5.6 m [12]. As for indoor positioning using reliability-augmented particle filter (RAPF). The average error of the experimental field established by this research was 1 m, and the 80% error was maintained within 1.8 m. The AMID [10] method obtained the mean error at 1.7 m with 90% error maintained within 5.6 m. The other method, called MVG [11], obtained a mean error of 2 m without the deviation rate data. As for the sensor array proposed by this research, combined with the RFMC, the average error was 0.73 m, and 90% of the error could be maintained within 1 m, showing that the proposed method has good accuracy and good stability; Table 4 shows the accuracy comparison.

## 5. Conclusions

At present, the magnetic field positioning technology mainly uses the magnetic field fingerprint comparison method to locate. By collecting the magnetic field information of each spatial point, we can build a magnetic field fingerprint map. When the user is positioning, the magnetic field measured by the sensor is matched with the magnetic field fingerprint map to identify the user’s location. However, since the magnetic field is easily affected by external magnetic fields and magnetic storms, which can lead to “local temporal-spatial variation”, this makes it difficult to construct a stable and accurate magnetic field fingerprint map. This research has built a magnetic sensor array to capture the spatial magnetic differentiation for indoor positioning. We propose a new magnetic indoor positioning method, RPNN fingerprint map classifier (RFMC), which uses a recurrent probabilistic neural network (RPNN) to perform real-time learning and inferencing. The magnetic sensor array can detect subtle magnetic anomalies and spatial variations to improve the stability and accuracy of magnetic field fingerprint maps, and the RFMC model is built for recognizing the magnetic field fingerprint. We can finally implement an embedded magnetic sensor array positioning system that was evaluated in an experimental environment. Our method can reduce the noise caused by the spatial-temporal variation of the magnetic field, thus greatly improve the indoor positioning accuracy, reaching an average positioning accuracy of 0.78 m.

As the demand for indoor positioning has attracted more attention, increasing research has proposed various methods of indoor positioning. However only a cost-less technique can be prevalent on the market. As we widely apply indoor positioning to commercial use, the development of our society can improve dramatically. A clear example is large shopping malls. Once shopping malls apply indoor positioning, it can not only reduce the chance of customers getting lost, but also collect their movement data. We can predict that, in the foreseeable future, our living can be more facilitated. This is also the reason we want to use magnetic fields as our major feature to build a model. As long as we can reduce the cost of implementing indoor positioning, more companies and industries will be willing to introduce this technology to their services.

## Figures and Tables

**Figure 1 sensors-21-05707-f001:**
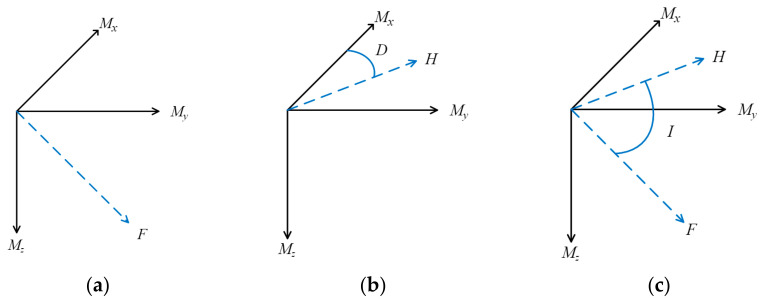
Geomagnetic field measurement: (**a**) geomagnetic field strength *F*; (**b**) geomagnetic field declination *D*; and (**c**) geomagnetic field inclination *I*.

**Figure 2 sensors-21-05707-f002:**
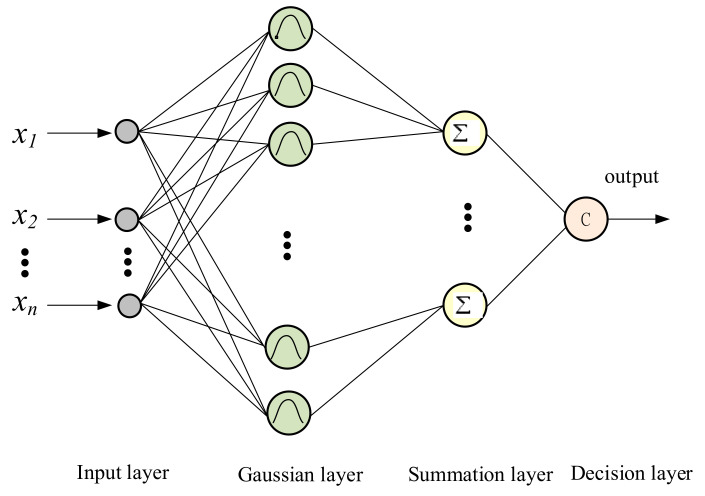
Architecture of a standard probabilistic neural network.

**Figure 3 sensors-21-05707-f003:**
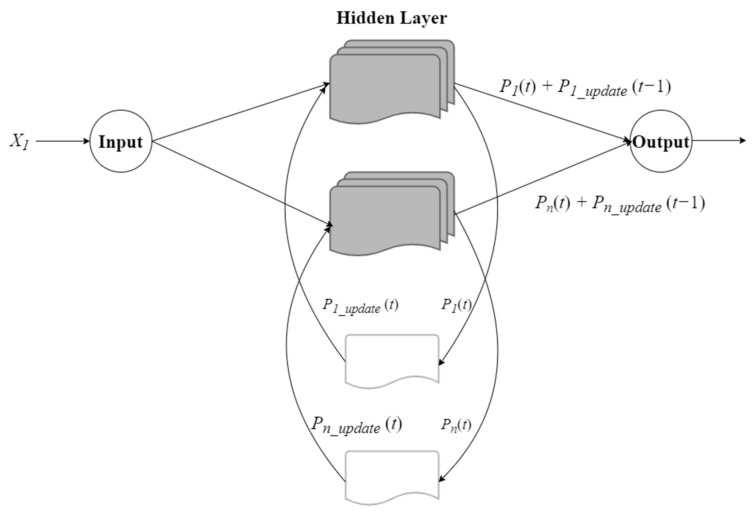
Recurrent probabilistic neural network model.

**Figure 4 sensors-21-05707-f004:**
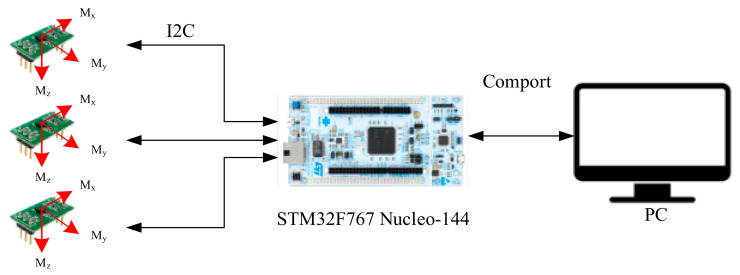
Embedded magnetic sensing array development platform.

**Figure 5 sensors-21-05707-f005:**
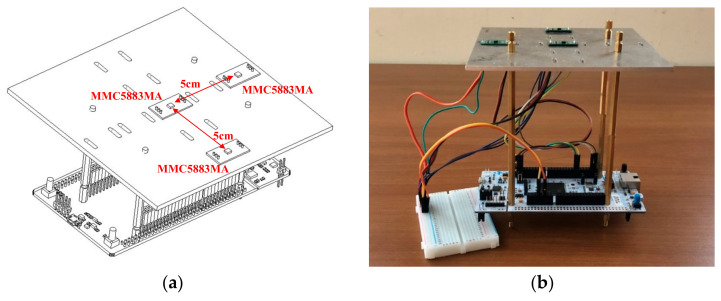
Magnetic sensor array indoor positioning system: (**a**) design and (**b**) prototyping.

**Figure 6 sensors-21-05707-f006:**
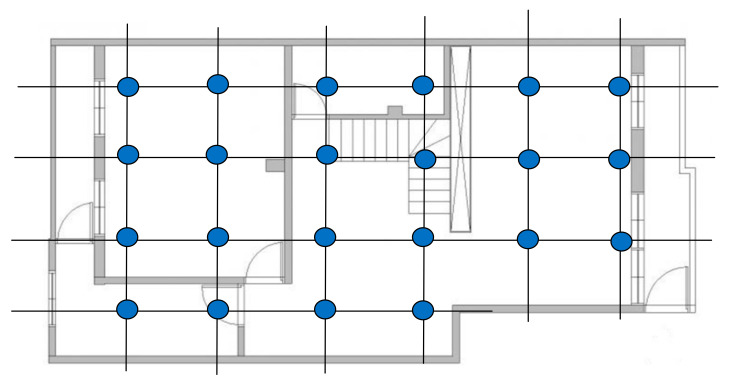
Divide the room into grids to create a magnetic field database.

**Figure 7 sensors-21-05707-f007:**
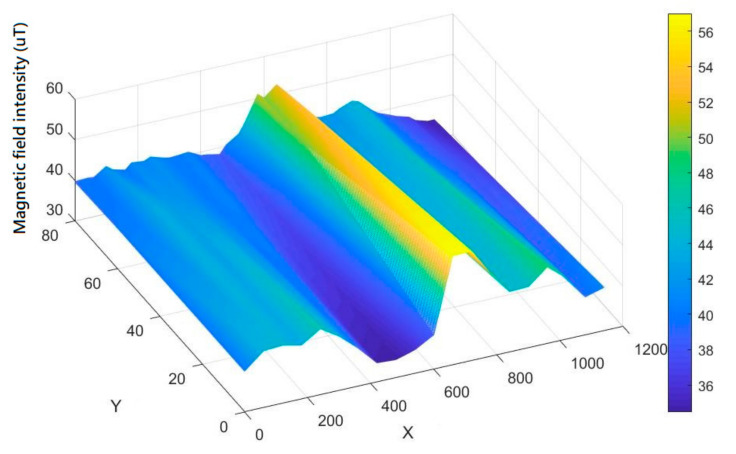
Schematic diagram of indoor magnetic field distribution.

**Figure 8 sensors-21-05707-f008:**
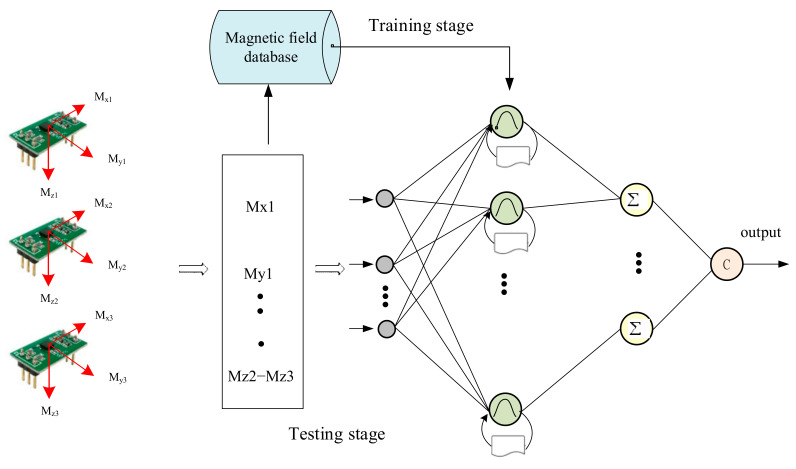
RPNN fingerprint map classifier (RFMC).

**Figure 9 sensors-21-05707-f009:**
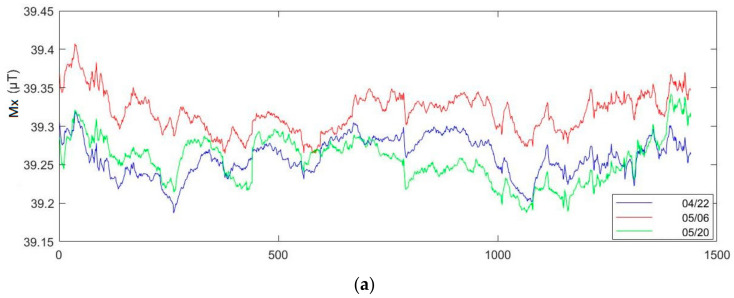
Changes in magnetic field strength collected 14 days apart by: (**a**) sensor X; (**b**) sensor Y; (**c**) sensor Z.

**Figure 10 sensors-21-05707-f010:**
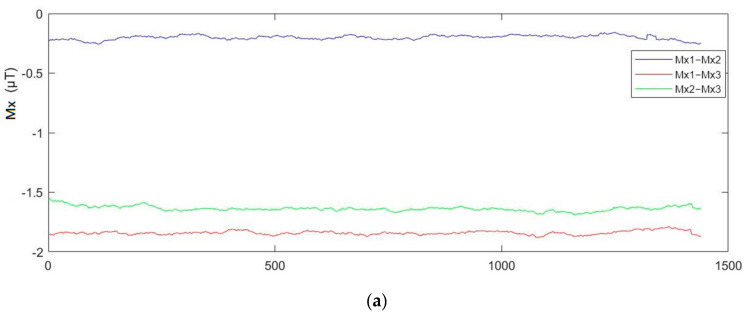
Differentiation of each magnetic signal component pair collected by: (**a**) sensor X; (**b**) sensor Y; (**c**) sensor Z.

**Figure 11 sensors-21-05707-f011:**
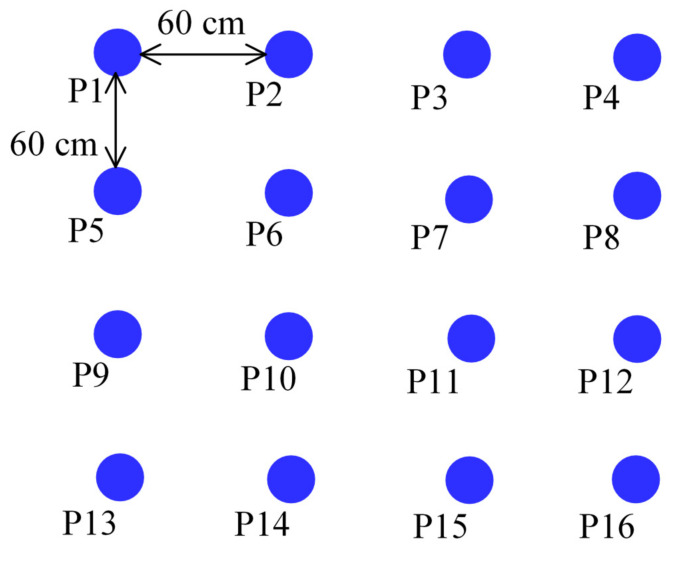
Sampling points arrangement.

**Figure 12 sensors-21-05707-f012:**
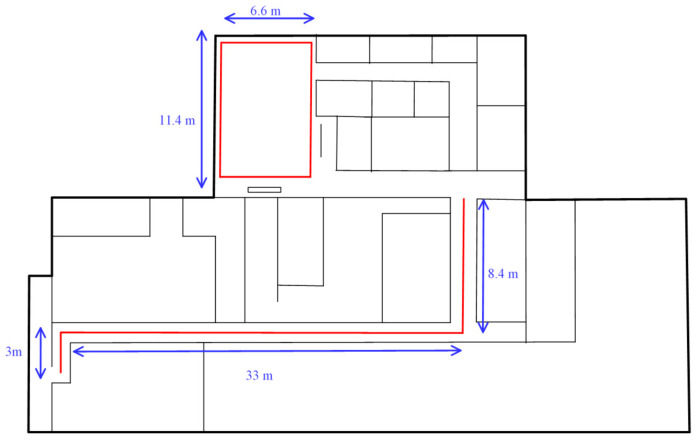
Floor plan of the first floor of the building.

**Figure 13 sensors-21-05707-f013:**
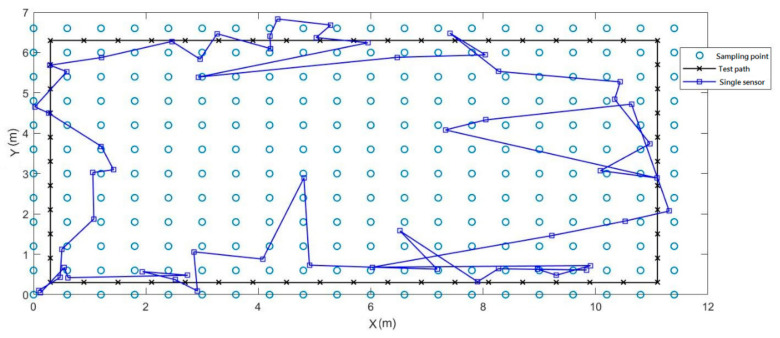
Indoor positioning experiment by single magnetic sensor.

**Figure 14 sensors-21-05707-f014:**
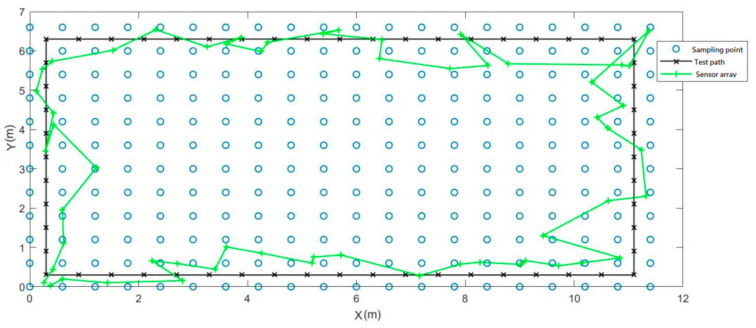
Indoor positioning experiment of magnetic sensor array.

**Figure 15 sensors-21-05707-f015:**
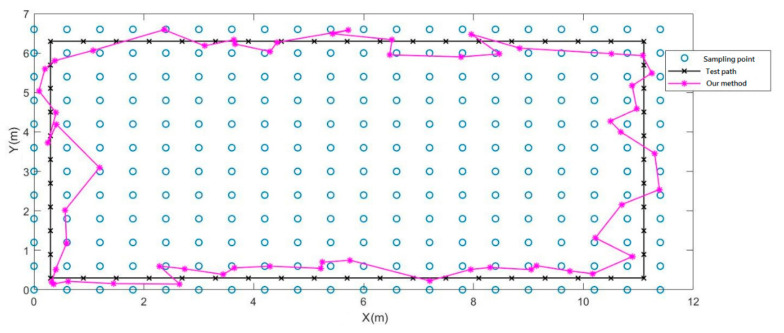
Indoor positioning experiment of sensor array with differentiation features.

**Table 1 sensors-21-05707-t001:** The differentiation calculation of the magnetic field strength of the grid sampling points in a small area.

	Sensor	x1 − x2	x1 − x3	x2 − x3	y1 − y2	y1 − y3	y2 − y3	z1 − z2	z1 − z3	z2 − z3
Position										
P1		−0.17	−0.78	−0.61	−3.65	−2.11	1.54	−25.87	−26.61	−0.74
P2		0.05	−0.71	−0.76	−3.98	−2.58	1.40	−27.44	−28.33	−0.88
P3		−0.42	−1.13	−0.71	−3.26	−2.41	0.85	−23.06	−23.99	−0.93
P4		−0.97	−1.54	−0.56	−1.78	−1.33	0.45	−13.43	−13.79	−0.36
P5		−0.39	−1.07	−0.68	−3.36	−2.11	1.25	−28.43	−29.73	−1.30
P6		−0.66	−1.03	−0.37	−4.41	−3.31	1.10	−32.59	−34.12	−1.53
P7		−0.54	−1.07	−0.53	−3.98	−3.12	0.86	−28.44	−29.56	−1.12
P8		−0.46	−1.17	−0.71	−2.27	−1.58	0.69	−16.14	−16.82	−0.68
P9		−0.78	−1.35	−0.57	−3.77	−2.68	1.09	−32.79	−33.96	−1.17
P10		−0.06	−1.06	−1.00	−5.14	−4.08	1.06	−42.14	−43.70	−1.56
P11		0.20	−0.88	−1.08	−4.63	−3.65	0.98	−37.19	−38.71	−1.52
P12		0.13	−0.86	−0.98	−2.87	−2.02	0.85	−23.05	−24.28	−1.23
P13		−1.66	−2.30	−0.64	−3.16	−3.03	0.13	−35.53	−36.46	−0.93
P14		−0.58	−1.56	−0.98	−4.77	−4.03	0.75	−44.67	−45.16	−0.49
P15		0.54	−0.69	−1.23	−5.33	−4.04	1.28	−43.25	−44.11	−0.86
P16		0.75	−0.62	−1.37	−3.85	−2.68	1.18	−32.17	−33.27	−1.10

**Table 2 sensors-21-05707-t002:** Performance comparison of different smoothing parameters for RFMC.

Field	Training Data	Test Data	Smoothing Parameter	Average Error (m)	Maximum Error (m)
hall	4800	480	0.3	0.77	1.37
			0.2	0.52	1.03
			0.1	0.48	1.16
corridor	1540	154	0.3	0.51	1.01
			0.2	0.38	0.84
			0.1	0.35	0.88

**Table 3 sensors-21-05707-t003:** Accuracy comparison of different configurations of RFMC.

	Average Error (m)	Maximum Error (m)
Single sensor	1.17	5.07
3-sensors array	0.83	1.91
3-sensors array + differentiation feature	0.73	1.73

**Table 4 sensors-21-05707-t004:** Accuracy comparison of different magnetic-based positioning methods.

Method of Positioning	Mean Error (m)	Deviation Rate (m)
CNN-based method [19]	1.7	90% of deviation < 5.6
RAPF method [12]	1	80% of deviation < 1.8
AMID method [10]	1.7	90% of deviation < 5.6
MVG method [11]	2	
Our method	0.73	90% of deviation < 1

## Data Availability

We have listed all the references we used in our research down below our manuscript.

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
