# Peer review of "Indoor Positioning Using Magnetic Fingerprint Map Captured by Magnetic Sensor Array"

_sensors, 2021, doi:10.3390/s21175707_

Round 1

Reviewer 1 Report

The paper is showing a vary interesting and promising approach in positioning technology. It will be very interesting to see the part of user location as an added value service that can support e.g. Emergency communication and NG112 positioning of persons.

The method looks very promising to assist and supplement other locations technology achieving event better accuracy.

It is important to show how the model works in concrete walls that some time is hard to overpass and if the magnetic sensor will have the same performance in these environments. While also solution in assisting living can be helpful and take advantage of the proposed solution a good starting point that Geofencing is considered in assisting live are the following DOIs. 10.1109/JIOT.2018.2885652 and 10.1109/MCC.2016.118 showing the practical use of the proposed solutions.

Additionally, some large sentences need to be dropped and a native speaker must go through the documents as some phrases are hard to follow.

The improve of the noise caused by the spatial-temporal variation needs to be justified compared to the lab environments

Author Response

Dear  reviewer:

Reviewer 2 Report

To me, the main concerns on this manuscript lie in the presentation, language, and the inadequate survey of existing papers.  However, the 0.78m accuracy is impressive to me and the method is reasonable. Therefore, I decided to give a chance to this manuscript.

  1. There are full of grammar errors in the manuscript, to name a few,
    a. By collecting the magnetic field information of each spatial point, then we can build a magnetic field fingerprint map.    (remove then) 
    b. However, since the magnetic field is easily affected by external magnetic fields and magnetic storms, which could lead to "local temporal-spatial variation" is generated, which makes ..., which makes it difficult...       (two "which" in one sentence, reconsider "is generated")
    c. We implement finally an embedded magnetic sensor array positioning system which is evaluated ... (we finally implement ?)
  2. Figure 2 describes the basic RNN structure, which is not necessary since this is almost known to all.
  3. The words in each section title or subsection title should be capitalized, e.g., Build the magnetic field database-->Magnetic Field Database Construction
  4. I have to say that the related work survey in this paper is very poor. Only 17 literatures are referred. Given that magnetic field signal based indoor localization is a hot topic that has tens of thousands of existing papers, this is obviously too narrow to survey existing advances in this field. To name a few,

    Magicol: Indoor Localization Using Pervasive Magnetic Field and Opportunistic WiFi Sensing, IEEE Journal of Selected Areas in Communications

    How feasible is the use of magnetic field alone for indoor positioning? ,  IPIN

    Using geomagnetic field for indoor positioning, Journal of Applied Geodesy

    Multiview and multimodal pervasive indoor localization, ACM Multimedia

    Design and evaluation of a wireless magnetic-based proximity detection platform for indoor applications. IPSN

    AMID: Accurate Magnetic Indoor Localization Using Deep Learning, Sensors

    They are all on magnetic based indoor localization.
  5. The method should compare with a few recent (after 2019) works.

Author Response

Dear reviewer:

Round 2

Reviewer 2 Report

Most of my concerns are addressed. The paper is indeed improved. I agree that the paper can be published now, but a careful check on the spelling and language is necessary.